# Large Simulated Future Changes in the Nitrate Radical Under the CMIP6 SSP Scenarios: Implications for Oxidation Chemistry

Scott Archer-Nicholls[1,*], Rachel Allen[1], Nathan L. Abraham[1,2], Paul T. Griffiths[1,2], Alex T. Archibald[1,2]

[1]Department of Chemistry, University of Cambridge, Cambridge, CB2 1EW, UK

[2]NCAS-Climate, University of Cambridge, Cambridge, CB2 1EW, UK

*Now at IT Services, University of Manchester, Manchester, M13 9PL, UK

*Correspondence to*: Alex T. Archibald (ata27@cam.ac.uk)

**Abstract.** The nitrate radical ($NO_3$) plays an important role in the chemistry of the lower troposphere, acting as the principle
oxidant during the night together with ozone. Previous model simulations suggest that the levels of $NO_3$ have increased dramatically since the pre-industrial. Here, we show projections of the evolution of the $NO_3$ radical from 1850-2100 using the UKESM1 Earth System model under the CMIP6 SSP scenarios. Our model results highlight diverse trajectories for $NO_3$, with some scenarios and regions undergoing rapid growth of $NO_3$ to unprecedented levels over the course of the 21st Century, and others seeing sharp declines. The local increases in $NO_3$ (up to 40 ppt above the pre-industrial base-line) are driven not only
by local changes in emissions of nitrogen oxides but have an important climate component, with $NO_3$ being favoured in warmer future climates. The changes in $NO_3$ lead to changes in the oxidation of important secondary organic aerosol precursors, with potential impacts to particulate matter pollution regionally and globally. This work highlights the potential for substantial future growth in $NO_3$ and the need to better understand the formation of SOA from $NO_3$ to accurately predict future air quality and climate implications.

## 1 Introduction

Whilst nitrogen is ubiquitous in the atmosphere, the majority of gaseous nitrogen-containing molecules are chemically inert. Reactive nitrogen species ($NO_y$) make up a much smaller fraction but encompass a diverse set of molecules that play a paramount role in the chemistry of the atmosphere. Nitrogen oxide (NO) and nitrogen dioxide ($NO_2$), collectively known as $NO_x$, are essential for the catalytic formation of ozone ($O_3$) in the troposphere, a key air pollutant and greenhouse gas (Monks
et al., 2015). The reaction of $NO_2$ with $O_3$ produces the nitrate radical ($NO_3$):

$$O_3 + NO_2 \rightarrow NO_3 + O_2 \qquad\qquad (1)$$

During the daytime, $NO_3$ can undergo rapid photolysis or reaction with NO resulting in a very short lifetime; typically in the order of seconds (Wayne et al., 1991). However, at nighttime, in regions of high $NO_x$, $NO_3$ is able to persist and become the

major oxidant of volatile organic compounds (VOCs) (the hydroxyl radical (OH) and ozone (O₃) dominate this oxidation during the daytime); acting as the most important oxidant during the night (Brown and Stutz, 2021). NO₃ also undergoes a reversible reaction with NO₂, forming a thermal equilibrium with dinitrogen pentoxide (N₂O₅):

$$NO_2 + NO_3 + M \rightleftharpoons N_2O_5 + M, \tag{2}$$

$$K_{eq} = \frac{[NO_2][NO_3]}{[N_2O_5]} \tag{3}$$

N₂O₅ has a short lifetime with respect to decomposition at typical atmospheric boundary layer temperatures. As temperature increases, the rate of decomposition of N₂O₅ increases resulting in a greater fraction of reactive nitrogen in the form of NO₃. This temperature dependence in the equilibrium between NO₂, NO₃ and N₂O₅ is further driven by the extreme temperature dependence in the formation of NO₃ through R1. Due to their tight chemical-coupling, NO₃ and N₂O₅ have been termed the N$_x$O$_y$ family (N$_x$O$_y$=NO₃+N₂O₅) (Stone et al., 2014). In this sense, sinks of N₂O₅ lead to corresponding indirect loss of NO₃. Loss of N$_x$O$_y$ has important implications for tropospheric ozone as it is an important nighttime reservoir of NO$_x$ and O$_x$ (O+O₃+NO₂+others) (Archibald et al., 2020a). In the daytime, any remaining N$_x$O$_y$ is converted back to NO$_x$, therefore nighttime sinks of N$_x$O$_y$ reduce the levels of NO$_x$ available to photochemically form ozone in the troposphere. The heterogeneous hydrolysis of N₂O₅, which occurs readily on aerosol surfaces, is one of the major nighttime sinks of reactive nitrogen in the troposphere (e.g., Riemer et al., 2003).

NO₃ reactions with alkenes proceed rapidly via addition of NO₃ to the double bond (Wayne et al., 1991, Brown and Stutz, 2012). As a result, the oxidation of biogenic volatile organic compounds (BVOCs), in particular terpenes that are emitted in vast quantities by the world's forests (Sakulyanontvittaya et al., 2008), is sensitive to the levels of NO₃ present (Ng et al., 2017). BVOC oxidation by NO₃ can lead to the formation of secondary organic aerosol (SOA) (Fry et al., 2009; Ng et al., 2017), fine aerosols that have implications for the planetary radiation budget and human health (Pöschl 2005). The extent that NO₃ initiated chemistry contributes to SOA formation (relative to O₃ and OH initiated production) depends on the molecular structure of the BVOC (and therefore their rate coefficients for reaction with NO₃), the rate of their emissions, and the levels of NO₃ present.

Ng et al. (2017) highlight that there are significant gaps in our understanding of the NO₃ initiated oxidation of BVOCs. BVOC oxidation chemistry has been studied for many decades (e.g., Brewer et al., 1984), however there are relatively few mechanistic studies of the products of NO₃ initiated oxidation (e.g., Boyd et al., 2015; Fry et al., 2009; Faxon et al., 2018, Ehn et al., 2017) compared to OH and O₃ (e.g., Atkinson and Arey, 2003; McGillen et al., 2020). Nonetheless there is mounting evidence that NO₃ oxidation could be a significant contributor to SOA formation. Kiendler-Scharr et al., (2016) highlight the ubiquity of organic nitrates in submicron aerosol collected in European nighttime urban conditions, whilst Hamilton et al., (2021) recently

found $NO_3$ oxidation of isoprene to be a significant source of SOA production in Beijing. Recent studies show highly oxidised molecules (HOMs), formed from the autooxidation and accretion of monoterpene oxidation products, are capable of nucleating new aerosol particles without sulphate seeds (Ehn et al., 2014, Trostl et al., 2016, Bianchi et al, 2019) with important implications for large-scale feedbacks between the climate and the biosphere (Scott et al., 2017, Gordon et al., 2017). Recently, Zhao et al. (2021) have shown that HOMs from $NO_3$ initiated oxidation of isoprene may contribute to a significant fraction of the isoprene SOA yield. But these complex processes are so far lacking from Earth system and global chemistry-transport models.

Oxidation of BVOCs is dependent on changes to the oxidant budget (with respect to OH, $O_3$ and $NO_3$) and to the changes in the emissions of BVOCs themselves. Our understanding of the processes which control emissions of BVOCs highlights that these have important dependences on climatic parameters including $CO_2$ concentrations (Arneth et al., 2007), land use, temperature, humidity and precipitation (Arneth et al., 2008). Whilst BVOC emissions models are reasonably well constrained in the present day, past and future trends are less certain, with some models indicating large increases in BVOC emissions in the future (Pacifico et al., 2012) and others small to none (Hantson et al., 2017).

Estimates of changes in the levels of $NO_3$ from the pre-industrial to the present day indicate significant increases, on the order of 100-1000% (Khan et al., 2015), driven by changes in emissions of $NO_x$. Formation of $NO_3$ is also dependent on ozone burdens which are likely to change in the future (Archibald et al., 2020a; Griffiths et al., 2021). In addition, the fraction of $N_xO_y$ in the form of $NO_3$ increases with temperature, while the main sinks are determined by the amount of $N_2O_5$ which decreases with temperature. Ng et al., (2017) postulated that $NO_3$ would increase with increasing $NO_x$ emissions but given the complex chemical processes at work $NO_3$ is likely to vary nonlinearly with $NO_x$ emissions. To our knowledge, no study has investigated how the $NO_3$ radical may change in the future under changing $NO_x$ emissions and a changing climate, or how these changes will affect BVOC oxidation and SOA formation given changing terpene emissions.

This study presents results on the historic and future evolution of $NO_3$ using simulations from 1850-2100 made with the United Kingdom Earth System model (UKESM1) (Sellar et al., 2019, Archibald et al., 2020b). The simulations, based on the Coupled Model Intercomparison Project Phase 6 (CMIP6) Historic and ScenarioMIP scenarios (O'Neill et al., 2016), indicate that there are substantial changes in $NO_3$ simulated in the future at both the global mean and regional level, depending on the emissions scenarios and resulting climate considered. Using these simulations we show that future trends in $NO_3$ have an impact on the regional oxidation of BVOCs and as a result the SOA budget and burden. Taking a particular focus on changes occurring over South Asia, a region with severe present day air quality issues and large variation in how emissions will evolve in this region under different scenarios, we highlight an important role for enhanced levels of $NO_3$ initiated BVOC oxidation in the future.

## 2 Methods

### 2.1 Description of UKESM1 model

The model simulations make use of the new U.K. Earth System Model (UKESM1) (Sellar et al., 2019). This is a fully coupled Earth system model using the Global Atmosphere 7.1/Global Land 7.0 (GA7.1/GL7.1; Walters et al., 2019). The dynamical model is coupled with the United Kingdom Chemistry and Aerosol model (UKCA), using the StratTrop mechanism for gas-phase chemistry (Archibald et al., 2020b) and the two-moment GLOMAP-mode aerosol scheme (Mulcahy et al., 2021). UKCA and UKESM1 are designed to quantify the response of the Earth system to forcings and simulate nascent feedbacks that exist in the coupled chemistry-aerosol-climate system. As such, pragmatic decisions have been made to ensure as complete a representation as possible of many processes are included in the model. For the chemistry of $N_2O_5$ this includes a fairly basic description of its chemistry in the gas phase and its loss to aerosol surfaces (heterogeneous chemistry) is simplified, with a fixed uptake coefficient being used ($\gamma=0.1$). Jones et al. (2021) suggest that the simplified treatment of heterogenous uptake of $N_2O_5$ leads to an overestimate of the loss of $N_2O_5$ based on comparison of modelled and measured $HNO_3$ across the USA. McDuffie et al. (2018) have determined a median $\gamma=0.0143$ (range: $24\times10^{-5}$ to 0.1751) based on constrained modelling of aircraft observations. Inclusion of this median uptake coeffecinet would be likely to lead to an increase in the atmospheric mixing ratio of $N_xO_y$ and thus our results can be seen as lower bounds on the potential changes in $[NO_3]$. UKCA does not include any in-particle chemistry for $N_2O_5$ and also lacks aerosol nitrate in the present implementation, similar to some other CMIP6 class models. The omission of aerosol nitrate is a weakness in the model but recent work including this process (Jones et al., 2021) suggests that the effects of it are small on $N_xO_y$.

The experimental setup is that of the simulations conducted as part of the Coupled Model Intercomparison Project Phase 6 (CMIP6; Eyring et al., 2016) DECK and ScenarioMIP experiments. Historical emissions are from the Community Emissions Data System (CEDS; Hoesly et al., 2018). Future emissions progress along one of four benchmark shared socioeconomic pathways (SSPs; Gidden et al., 2019). The analysis focuses on four representative pathways, SSP1-2.6, SSP2-4.5, SSP3-7.0 and SSP5-8.5. Each SSP has different assumptions about the amount of emissions of air pollutant precursors and climate forcers (reference changes in emissions of $NO_x$ under the scenarios (Gidden et al., 2019)). We perform our analyses focused on 5 year periods (Preindustrial, PI: 1850-54; Present day, PD: 2010-14; end of century 2090-94). These were conducted by performing new simulations using re-start files from the start dumps from the core CMIP6 simulations contributed by UKESM1. It was necessary to re-run the CMIP6 simulations with UKESM1 in order to (i) provide high temporal resolution output with additional diagnostics for the $NO_3$ reactions which did not exist in the CMIP6 runs, needed to assess the oxidation of Monoterp and isoprene and (ii) include a correction to the $NO_3$+Monoterp reaction rate coefficient.

### 2.2 Description of relevant chemical reactions

The representation of BVOC chemistry in the Strat-Trop chemical mechanism used in the UKESM1 model is similar to other Earth system models in being very simplified. Isoprene is treated as an individual compound undergoing reactions with OH, $O_3$ and $NO_3$, while monoterpenes are represented with a surrogate species, Monoterp, which can undergo oxidation via OH, $NO_3$ or $O_3$ with rate coefficients equal to the equivalent oxidation reactions of α-pinene to form an inert species, SEC_ORG, which irreversibly condenses to form SOA (Mann et al., 2010). This simplification has been shown to broadly capture the relation between BVOC emissions, SOA formation and the climate impacts well (Scott et al., 2017, Mulcahy et al., 2020).

All monoterpenes are represented by the "Monoterp" species, which undergoes oxidation via OH, $O_3$ and $NO_3$ to form the "SEC_ORG" species, which in turn undergoes irreversible condensation to form secondary organic aerosol (SOA):

$$\text{Monoterp} + \text{OH} \rightarrow F * \text{SEC\_ORG}; \ k_{OH}$$
$$\text{Monoterp} + O_3 \rightarrow F * \text{SEC\_ORG}; \ k_{O3}$$
$$\text{Monoterp} + NO_3 \rightarrow F * \text{SEC\_ORG}; \ k_{NO3}$$

Where F is some factor between 0 and 1 representing the yield of SOA production from MONOTERP. The assumed yield of SEC_ORG from Monoterp is 0.13, but is doubled to 0.26 in order to account for the missing production from isoprene oxidation (Mann et al., 2010, Mulcahy et al., 2020). The rate coefficients $k_{OH}$, $k_{O_3}$ and $k_{NO_3}$ are equal to the rate coefficients for the reactions between α-pinene and OH, $O_3$ and $NO_3$ respectively from Atkinson et al. (1989). Due to an error, the reaction rate coefficient activation energy for the $NO_3$+Monoterp reaction was incorrect in the original version of UKESM1 and has been corrected for these experiments. The rate coefficient used in all previous studies using StratTrop with GLOMAP aerosol was:

$$k_{NO_3} = 1.19 \times 10^{-12} \times e^{(925/T)} \text{ cm}^3 \text{ molecule}^{-1} \text{ s}^{-1}$$

Whereas the correct form is:

$$k^*_{NO_3} = 1.19 \times 10^{-12} \times e^{(490/T)} \text{ cm}^3 \text{ molecule}^{-1} \text{ s}^{-1}$$

At *T=298K*, this difference in activation energy results in $k_{NO_3}$ being 4.3 times faster than $k^*{}_{NO_3}$. This correction results in a smaller fraction of Monoterp being oxidised by $NO_3$ compared to OH and $O_3$, partly compensated by an increase in $NO_3$ burden (see Supplement further details).

## 3 Results

### 3.1 Comparison of Present day modelled and observed $NO_3$.

Our analysis begins by evaluating the performance of UKESM1 against aircraft observations and pre-compiled ground-based observations presented in Khan et al. (2015). Figure 1 shows the comparison of UKESM1 modelled vertical profiles of $O_3$,

NO$_2$, NO$_3$ and NO$_3$:N$_2$O$_5$ against observations made in the Eastern UK, over the North Sea, as part of the RONOCO summer campaign that took place in July 2010 (Stone et al., 2014). UKESM1 output is not constrained by meteorological forcing so we sample the model during July 2010-2014, outputting model data once every 27 hours and sampling data between 21:00 and 03:00 local time to generate a nighttime climatology for July, which is plotted in orange in Figure 1 with the error bars reflecting the variability (1 σ) in the model climatology. In general the model simulates the vertical profile of the RONOCO observations well. There is a positive bias in modelled O$_3$ and a negative bias in modelled NO$_2$ (consistent with previous analyses with UKCA the chemistry component of UKESM1 (e.g. Archibald et al., 2020b)). These biases partly offset each other; the vertical profile of the rate of production of NO$_3$ [Eq. 1] is in good qualitative agreement between the model and observations, with maxima found above the surface below 1 km, but the absolute magnitude of the NO$_3$ production rate is underestimated in the model by a factor of ~ 2 in this region. One reason for the model disagreement is likely the resolution, whereby the UKESM1 model is unable to simulate the fine plumes of NO$_2$ that drive NO$_3$ production that were observed during the RONOCO campaign (Stone et al., 2014). In addition to the poor horizontal resolution models like UKESM1 also suffer in biases in vertical resolution and mixing. The simulation of boundary layer height in models like UKESM1 is difficult and Figure 1 suggests that the model boundary layer height is much lower than the observations. This would make sense if in the model there is a significant land fraction in the grid boxed being analysed, as is the case.

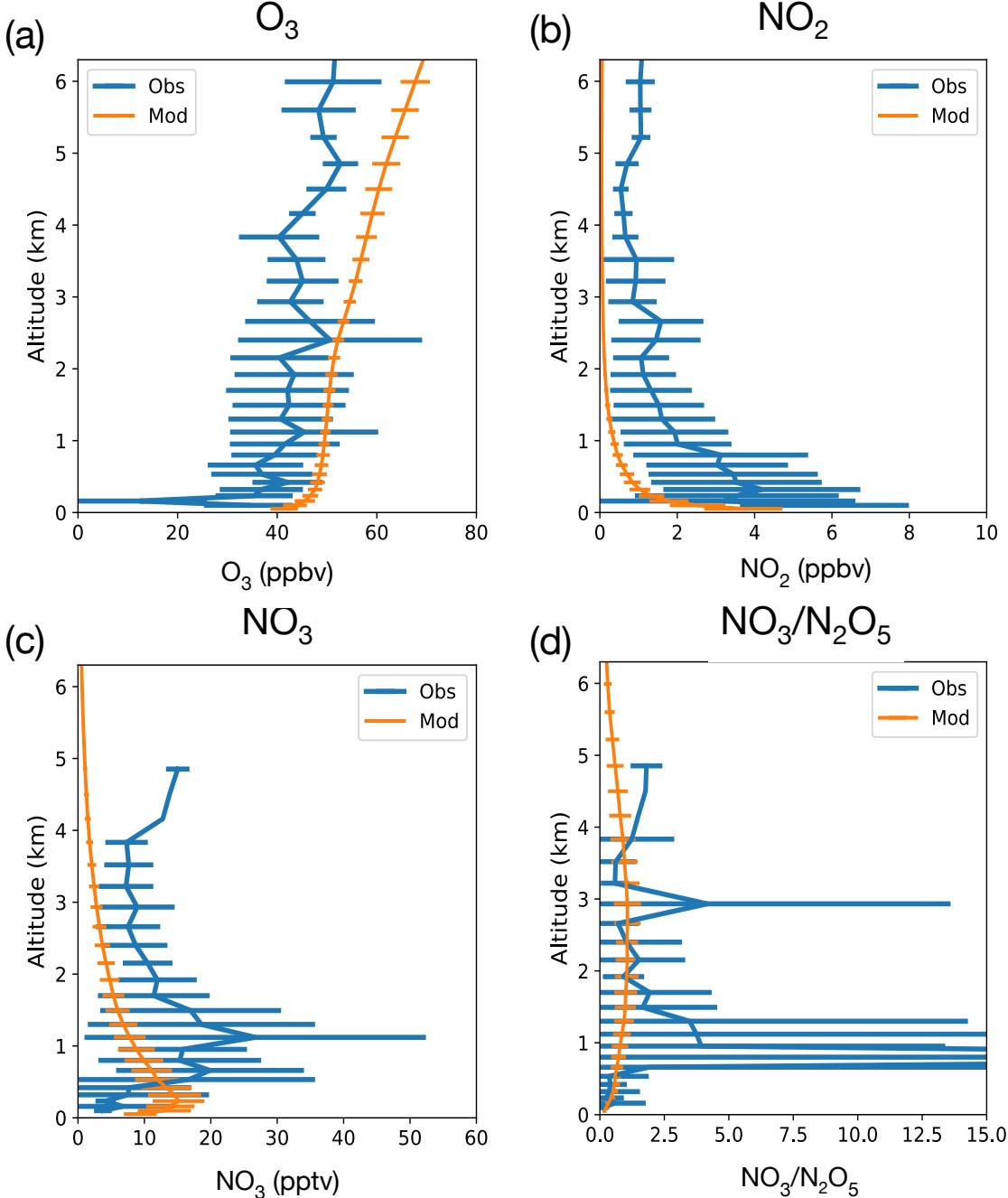

**Figure 1.Comparison of vertical profile of night-time concentrations of O₃ (a), NO₂ (b), NO₃ (c) and NO₃:N₂O₅ ratio (d) between RONOCO July 2010 flight campaign and 2010-2014 July nighttime values from UKESM1 model simulations from grid cells corresponding to flight regions. Lines show means and error bars show standard deviation.**

Table 1 compares results from the Historical simulation of UKESM1 against the STOCHEM-CRI model data from Khan et al. (2015) and a synthesis of surface observations; see Khan et al. (2015) for details and references for observational data. Our model data are averages from the 2010-14 period corresponding to the months of each campaign and are in good qualitative and quantitative agreement with the observations (which come from a range of different techniques and cover a range of different dates over the period c.a. 1990s-2010s, see Khan et al. (2015) for details). The largest disagreement in Table 1 is at Guangzhou, China, where both models predict much lower levels of $NO_3$ than were observed using a long-path DOAS instrument (Li et al., 2012). Despite being of low horizontal and vertical resolution, and in light of the caveats already discussed through the analysis of Figure 1, we find that UKESM1 reproduces well current observations of $NO_3$ suggesting it is a suitable tool for simulating past and future changes in the $NO_3$ burden and distribution.

**Table 1. Monthly average values from of $NO_3$ mixing ratios (ppt) from UKESM1 compared with review of $NO_3$ measurements and STOCHEM model output adapted from Table 3 in Khan et al., (2015).**

| Site | Start | Observed (ppt) | STOCHEM (ppt) | UKESM1 (ppt) |
|------|-------|----------------|---------------|--------------|
| Lindenberg | Feb-Sep 1998 | 4.6 | 6.9 | 8.0 |
| Tanus | May 2008 | 30±20 | 6.9 | 6.9 |
| Jerusalem | Jul 2005–Sep 2007 | 27±43 | 9.2 | 11.2 |
| Houston | Aug-Sep 2006 | 0-149 | 3.3 | 11.0 |
| Shanghai | Aug-Oct 2011 | 16±9 | 16.6 | 8.4 |
| Guangzhou | Jul 2006 | 21.8±1.8 | 4.7 | 4.4 |
| Izu-Oshima | Jun 2004 | 3 | 4.4 | 7.5 |
| Mace Head | Jul-Aug 1996 | 5 | 3.8 | 3.1 |
| Canary Islands | May 1994 | 8±3 | 5 | 5.7 |
| Finokalia | Jun 2001– Sep 2003 | 4.2±2.3 | 16 | 16.9 |
| Weybourne | Oct-Nov 2004 | 6 | 5.5 | 4.2 |
| East Point | Jul-Aug 2005 | 13.1 | 1.6 | 1.4 |
| Schauinsland | Aug 1990 | 5.8 | 12 | 6.3 |

## 3.2 Modelled Past and Future Changes in $O_3$, $NO_x$ and $NO_3$

Our analysis focuses on five year periods from the simulations, comparing present day (PD; 2010-2014) and preindustrial (PI; 1850-1854) periods with end of century (2090-2094) predictions under four shared socioeconomic pathways (SSPs) as used for CMIP6. Data for these five year periods were collected by rerunning the UKESM1 CMIP6 simulations with additional diagnostics for analysis and corrected $NO_3$+MONOTERP reaction rate coefficient (see Methods section for details). The future emission pathways we studied, described in more detail in Gidden et al. (2017), are in short: SSP1-2.6, a 2˚C world with a focus on sustainability, economic growth and connectivity and low population growth driven by low-carbon technology and energy efficiency; SSP2-4.5, a middle-of-the-road scenario with slower convergence of income levels, greater population growth and reliance on fossil fuels for further into the century; SSP3-7.0, a regional rivalry scenario with increased inequality and high population growth in low-middle income countries; and SSP5-8.5, similar to SSP1 in terms of population and economic growth but driven by increasing unabated use of energy and fossil fuels. Although SSP5-8.5 leads to the greatest increase in long-lived greenhouse gas emissions, and therefore global warming, by the end of the century, SSP3-7.0 has the highest emissions of short-lived air pollutants (Rao et al., 2017). Emissions of BVOCs also change between these scenarios as they are interactive in UKESM1 (Sellar et al., 2019). BVOC emissions are higher in the PI compared to PD due to changing land use, reducing forested area over the 20[th] century. In the future SSP scenarios, BVOC emissions are predicted to increase as temperatures increase, in spite of the increasing $CO_2$ levels (which, in the absence of other changes, would cause BVOC emissions to decrease (Arneth et al., 2008)).

The results from the UKESM1 simulations show that large changes to monthly average $O_3$, $NO_2$ and $NO_3$ levels have taken place in the lowest km of the atmosphere from PI to PD (Fig. 2 a-c). The region of the lowest km is analysed here, rather than the surface level, because nighttime $O_3$ and $NO_3$ levels tend to be low at the surface due to rapid reaction with fresh emissions of NO and the low nighttime boundary layer. In both observations and the model, peak $NO_3$ tends to occur a few hundred meters above the surface (e.g. Fish et al., 1999).

The main driver for the changes between the PI and PD periods is attributed to the increase in anthropogenic emissions, with emissions of $NO_x$ increasing by orders of magnitude over this period. The largest increases in $NO_2$ and $O_3$ occur over populated regions (Fig. 2 (a, b)). The changes in $O_3$ simulated with UKESM1 are in good agreement with other CMIP6 models (Griffiths et al., 2021) and show an increase in the tropospheric burden of around 40% from the PI-PD. $NO_3$ is similarly much higher in the PD compared to PI (Fig. 2 (c)), an increase in tropospheric burden of approximately 75%, in good agreement with previous global modelling studies (Khan et al., 2015) and with observations (see Section 2.1). To first order, $NO_3$ is proportional to $O_3 \times NO_2$ as these species drive its production, but there are some important deviations from this relationship. Firstly, concentrations are lower at high latitudes because colder climates mean more $NO_3$ is partitioned into $N_2O_5$. Changes in $NO_3$ are also supressed in regions with high BVOC emissions, such as South-East USA, South America and South-East Asia/South

China, due to the high reactivity with these species. Greatest increases are over the middle East and Indian subcontinent – regions with high local temperatures, NOx emissions and $O_3$, as well as relatively low BVOC emissions.


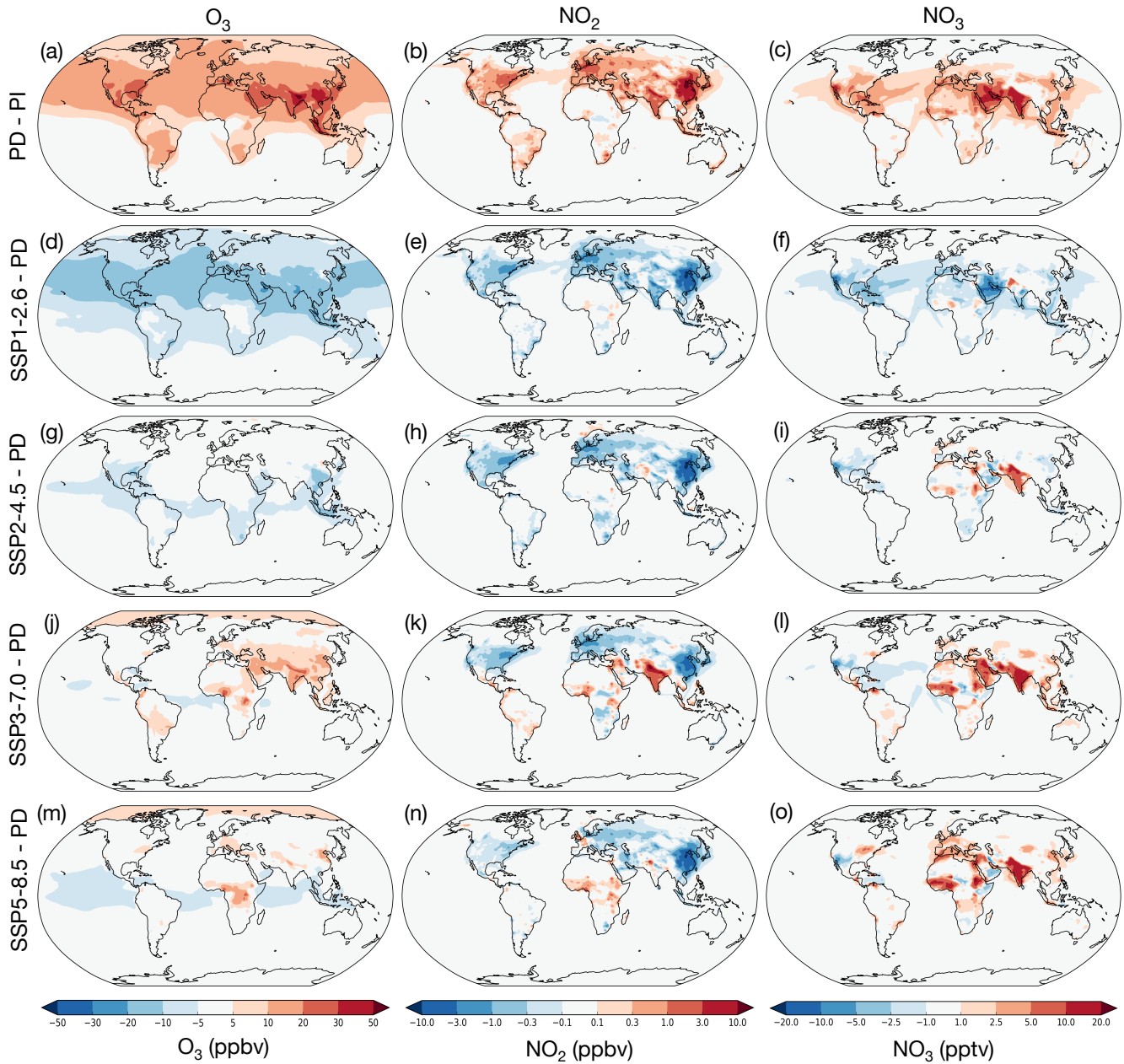

Figure 2. Changes in $O_3$, $NO_2$ and $NO_3$ mixing ratios averaged over lowest 1km of the atmosphere. Showing difference between Present Day (2010-2014) and Preindustrial (1850-1854) (a-c), SSP1-2.6 (2090-2094) – PD (d-f), SSP2-4.5 (2090-2094) – PD (g-i), SSP3-7.0 (2090-2094) – PD (j-l), and SSP5-8.5 (2090-2094) – PD (m-o).

The simulated future changes of $O_3$, $NO_2$ and $NO_3$ in the lowest 1km depend greatly on the future SSP scenario. Focusing on the change between the future and the PD, the SSP1-2.6 scenario shows large reductions in $O_3$, $NO_2$ and $NO_3$ across the world, especially in the Northern Hemisphere (Fig. 2 (d-f)). These reductions are similar to but smaller than the increases in $NO_3$ from the PI to PD, leading to a tropospheric $NO_3$ burden under SSP1-2.6 that is 26% lower than the PD burden. In SSP2-4.5 (Fig. 2 (g-i)) $NO_2$ decreases in most regions, although to a lesser degree in some regions such as India. However, reductions in $O_3$ are

muted, with similar concentrations across most of the Northern Hemisphere; likely due to competing trends from changes to stratosphere-to-troposphere transport of ozone (driven by increases in the Brewer-Dobson circulation under climate change), lower $NO_x$ emissions (decreasing ozone), and higher temperatures and BVOC emissions (increasing ozone) (e.g., Archibald et al., 2020a). Changes in $NO_3$ are spatially variable, with small reductions in North America and China but increasing levels in Northern Europe, India and parts of Africa. In the SSP3-7.0 scenario (Fig. 2, (j-l)), which simulates the greatest increase in

emissions of short-lived pollutants, near-surface $O_3$ is predicted to increase over most populated regions, particularly in South Asia and West Africa. In contrast, $NO_2$ shows diverging trends, decreasing in North America, Europe and China but increasing considerably in India, the middle East and West Africa. Over most parts of the land and oceans $NO_3$ is predicted to not change significantly but specific regions show dramatic increases: by over 10 ppt in West Africa, the middle East and India (Fig. 2 (l)). These large increases in $NO_3$ are of a similar order to the PI-PD changes shown in Fig. 2 (c), effectively doubling near-

surface $NO_3$ concentrations in the future over many populated regions, whilst $O_3$ increases are only in the order of 50% greater than the PD. In SSP5-8.5 (Fig. 2 (m-o)), the warmest future scenario, the increases in $O_3$ in Asia are small and $NO_2$ is predicted to remain similar to present day in India whilst decreasing significantly in China. However, there is still a considerable increase in $NO_3$ over India as well as West Africa and Europe predicted under the SSP5-8.5 scenario (in the order of 10-20 ppt increases above PD levels). These variations in trends between scenarios show that it is not sufficient to assume that lower $NO_x$ emissions

in the future will result in reduced $NO_3$. Rather, it is a complicated outcome also depending on other changes in the chemical environment and climate.

To investigate the role of changes in climate on $NO_3$, through changes in temperature over the lowest 1km of the atmosphere, the natural logarithm of $K_{eq}$ (Equation 3) is plotted against the corresponding temperature values in Figure 3, using data from

each of the scenarios. This shows the strong dependence of $K_{eq}$, in accordance with the Van't Hoff relationship. As the climate warms the thermal equilibrium shifts such that more $N_xO_y$ is found as $NO_3$, increasing the amount of $NO_3$ available for oxidation and reducing the sink due to the $N_2O_5$ heterogeneous reactions. UKESM1 simulates levels of warming by the end of the 21st century that are at the higher end of the CMIP6 multi model mean but within the ensemble spread (Tebaldi et al., 2021).


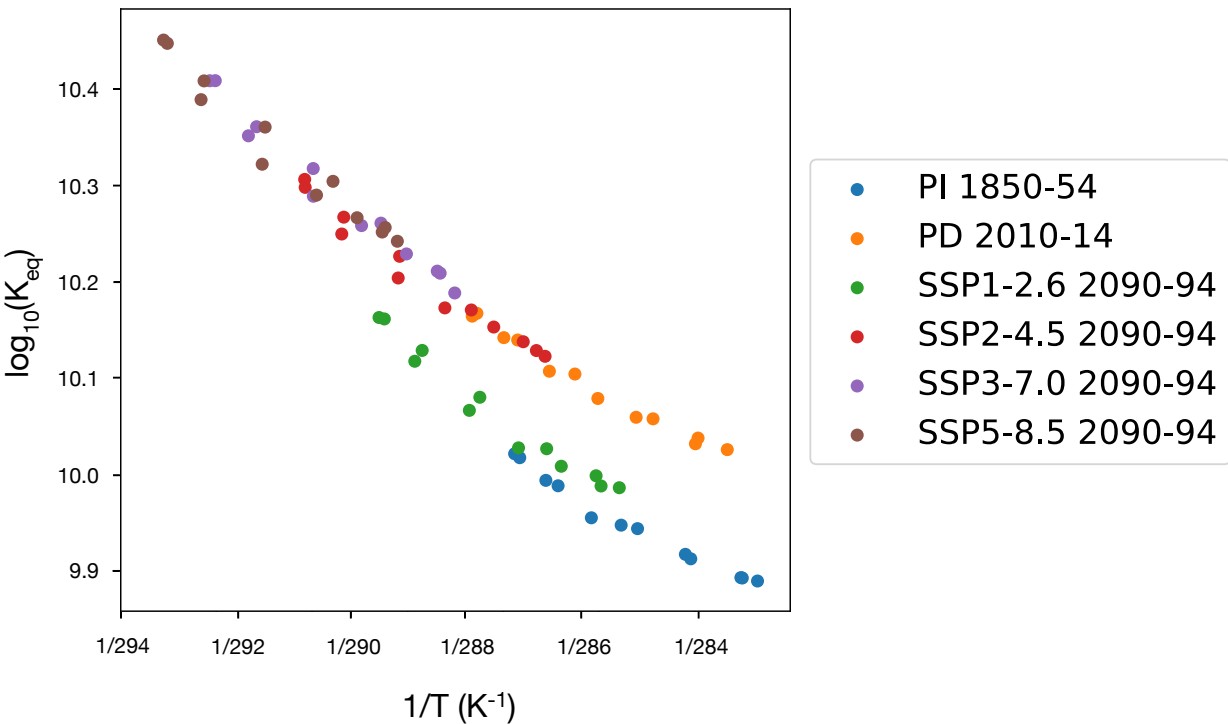

**Figure 3. Van't Hoff isochore of the thermal equilibrium between NO₃ and N₂O₅ (a) and change in temperature in UKESM1 model simulations from 1850 to 2100 in four representative SSP scenarios (b) changes in temperature from historical simulation from 1850 to 2014, and along SSP scenarios from 2014 to 2100.**


### 3.3 Changes to BVOC oxidation

Figure 4 shows vertically integrated modelled oxidation fluxes for isoprene and MONOTERP (the lumped monoterpene species in UKESM1) averaged across the globe and over Southern Asia (SA; defined as 5°N, 50°E to 35°N 95°E, based on the source-receptor region used by the Task Force on Hemispheric Transport of Air Pollution, TF HTAP, www.htap.org . Plots
for the other TF HTAP regions are included in Supplement). The modelled oxidation flux is calculated at each model chemistry timestep (1 hour),  thereby taking into account the concentrations of the BVOC, oxidants and the temperature dependent rate coefficients for their reactions, then averaged over each month for analysis. The fluxes are further averaged over the lowest 1 km of the atmosphere to reflect boundary layer oxidation in Figure 4. In the PI atmosphere, NO₃ initiated oxidation accounts for less than 1% of isoprene and 4% of MONOTERP, due to the low NOₓ emissions in this period (Fig. 2b). This fraction of
oxidation increases to 3% and 13% for isoprene and MONOTERP respectively for present day (i.e. a factor of 3 increase). UKESM1 predicts that Monoterpene emissions increase in a warming climate in all scenarios, particularly in SSP3-7.0 and

SSP5-8.5. Across the whole troposphere, the relative amount via NO₃ stays the same in SSP3-7.0 relative to PD, although the absolute amount increases as BVOC emissions increase. However, over the South Asian region the fraction of MONOTERP oxidised by NO₃ increases from 13% to 15%. In SSP1-2.6, MONOTERP oxidation fraction by NO₃ decreases in all regions

evaluated.

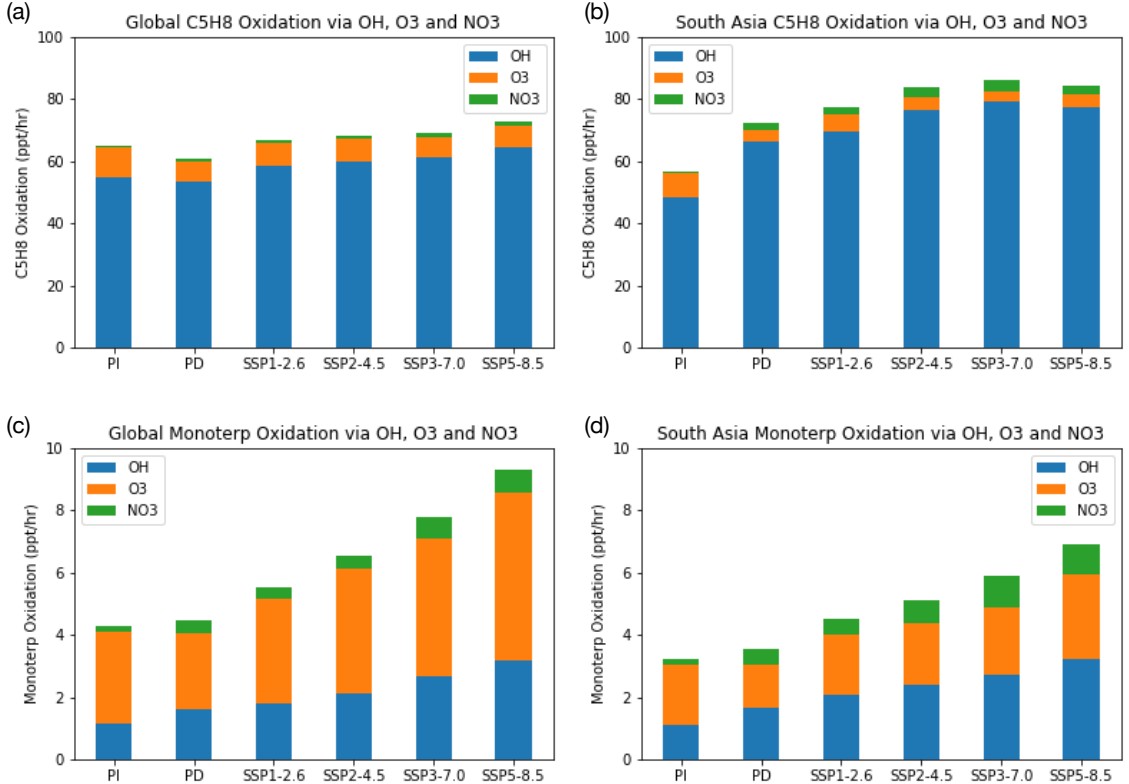

**Fig. 4 Average oxidation rates in lower 1km of atmosphere for isoprene (a and b) and MONOTERP (c and d), averaged across the whole globe (a, c) and over South Asia region (b, d).**


Focusing on MONOTERP oxidation, which in UKESM1 leads to the formation of SOA, Figure 5 shows the vertical profiles of the oxidation of this compound under PD conditions (panels (a) and (c)) and the SSP3-7.0 future scenario (panels (b) and (d)). Similar figures for the other SSP scenarios are included in the Supplement. As with Figure 4, Figure 5 shows that the future scenario results in an increase in oxidation of MONOTERP (an increase in SOA production in the model) for all

oxidants. The vertical profiles show a sharp drop in the oxidation flux away from the surface as a result of the relatively short lifetime of the MONOTERP species, which is emitted in the model at the surface and decays away in the vertical rapidly. Focusing on the near surface, where the rate of SOA production via MONOTERP oxidation is highest, Figure 5 shows that over Southern Asia the UKESM1 model predicts an increase in near surface NO₃ oxidation of MONOTERP of a factor of 2

under the SSP3-7.0 scenario. This implies potential for enhanced production of organic nitrogen containing aerosols under
future climate and emissions scenarios, compounds whose health and climatic effects have been hitherto less well studied but
observational evidence suggests are ubiquitous in the atmosphere (Kiendler-Scharr et al., 2016).

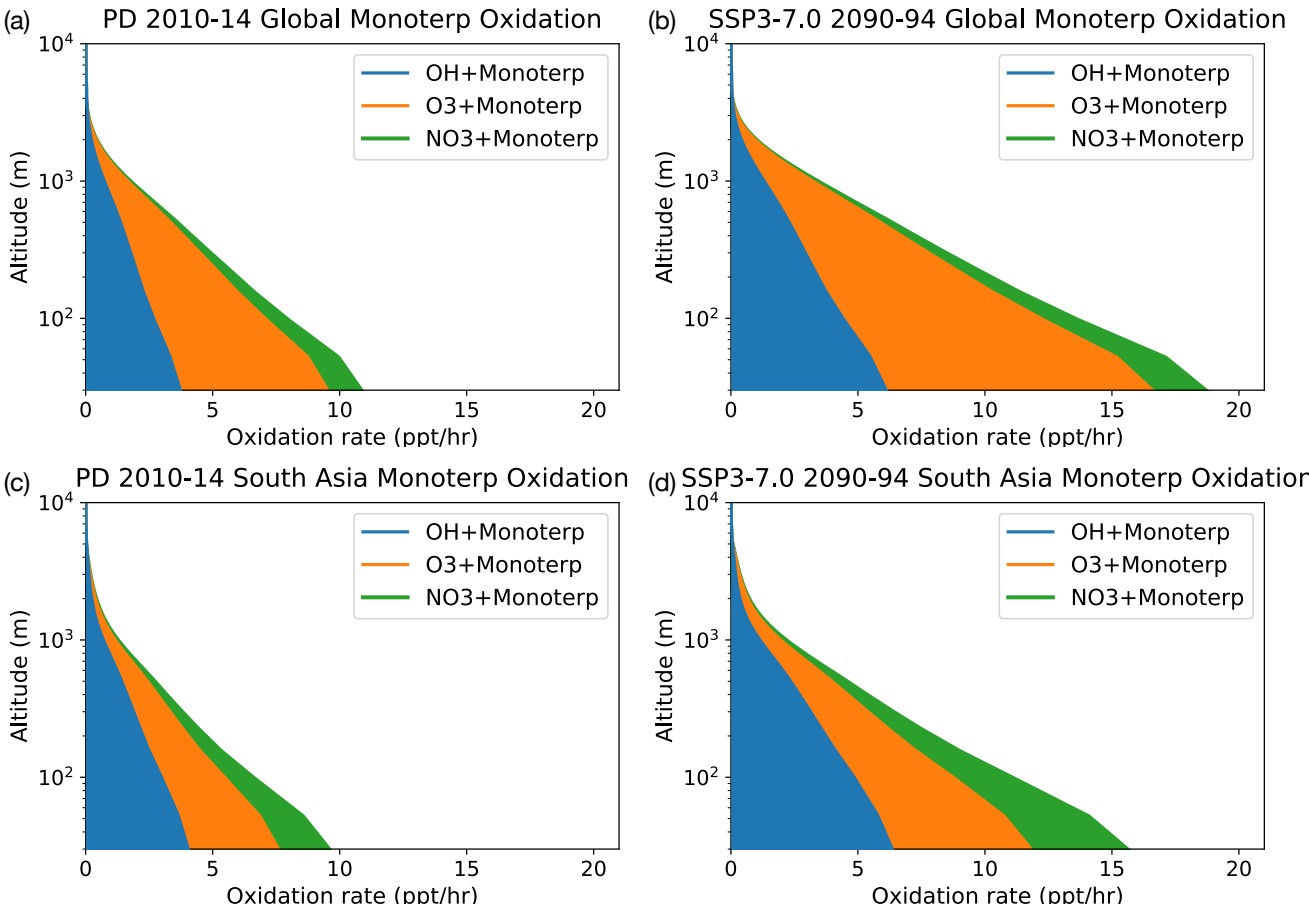

**Fig. 5. Vertical profile of MONOTERP oxidation via OH, O$_3$ and NO$_3$. Panels (a) and (c) show the model calculated present day global and South Asian oxidation fluxes respectively. Panels (b) and (d) show these fluxes at the end of the 21$^{st}$ Century under the SSP3-7.0 scenario. The future scenario results in increases in all oxidation fluxes, with the NO$_3$ initiated oxidation flux increasing by a factor of 2 compared with the present day oxidation rates in South Asia.**

## 4 Discussion

Whilst we have demonstrated large potential changes in NO$_3$ in the future under the SSP scenarios, we acknowledge that there
are some limitations to the state-of-the-art CMIP6 models, like UKESM-1. Faithful simulation of NO$_3$ requires models which
capture a wide range of processes as discussed in Ng et al. (2017). These include the ability to simulate faithfully the diurnal
variability of the boundary layer, which has been shown to be a challenge for these types of models (for example in East Asia

Yue et al. (2021) have shown that CMIP6 models can not reproduce observed trends in the boundary layer height); Simulating the diurnal emissions of important $NO_3$ sinks, like isoprene (Cao et al., 2021) – which almost all interactive chemistry CMIP6

models do – and $NO_3$ sources, like $NO_x$, which no CMIP6 models do. Further work is required to improve the representation of these key emissions sources and processes in global chemistry-climate models. In addition, the simplistic treatment of SOA in UKESM1 prevents us being able to explore how the composition of SOA may change under the changing climate and emission scenarios that have been explored for CMIP6. Many previous studies have highlighted how the formation of SOA is highly sensitive to the conditions it is being formed under and how the composition of SOA will change concomitantly (e.g.,

Hoyle et al., 2011; Pye et al., 2010; Schwantes et al., 2019). These feedbacks between changes in SOA formation mechanism and SOA composition are being explored through future work in UKESM.

## 5 Conclusions

In this work we have demonstrated the potential pathways for future evolution of $NO_3$, the most important oxidant at night, in the lower troposphere. Analysis of our model simulations against historic measurements highlights that in spite of the relatively

low resolution of the UKESM1 model, it is able to capture the magnitude and variability of observations of $NO_3$ and its precursors ($O_3$ and $NO_2$). We have assessed four different future scenarios which span a wide range of possible $NO_x$ and VOC emission pathways, and levels of climate change, simulated using the UKESM1 Earth system model. This is, to our knowledge, the first assessment of future simulations of $NO_3$ and this work highlights the potential for significant increases in this major night-time oxidant. In particularly we have shown that, depending on the emissions scenario, regions of Southern Asia are an

area of particular interest with potential increases in $NO_3$ to unprecedented levels; more than double their present day values. We also demonstrate the importance of climate change on $NO_3$, and show that the ratio of $NO_3:N_2O_5$ is predicted to increase under the higher climate forcing future scenarios. The impacts of an increase in $NO_3$ dominated BVOC oxidation are, as yet, uncertain. An increasing body of literature is examining the mechanistic pathways through which BVOCs and $NO_3$ react and the impacts of BVOC+$NO_3$ derived SOA. He et al. (2021) have shown through detailed laboratory experiments using cavity

enhanced absorption spectroscopy that hat some of the organic nitrates in BVOC+$NO_3$ derived SOA may serve as atmosphere-stable NOx sinks, or reservoirs, and will absorb and scatter incoming solar radiation during the daytime leading to an anthropogenic radiative forcing component (given that $NO_3$ is primarily an anthropogenic species).

The potential increases in $NO_3$ are shown to be important in the context of enhancing the oxidation of BVOCs in the

atmosphere. Due to the poorer understanding of SOA formation from $NO_3$ oxidation (Ng et al., 2017) than from OH or $O_3$ oxidation, these findings highlight the need for further lab and field studies to better understand the SOA forming potential of BVOC oxidation initiated via $NO_3$ and the incorporation of these data into global chemistry and Earth system models. Our simulated changes in $NO_3$ are likely dependent on (1) model structure, more work investigating the impacts of vertical and horizontal resolution is required. Our estimates are that low resolution Earth system models like UKESM1 will underestimate

the peak concentrations of $NO_3$ owing to poor vertical mixing; (2) emissions, including soil NOx emissions, which are highly uncertain still, and the diurnal and hebdomadal variation in anthropogenic ones – as with most CMIP6 models the emissions in UKESM1 are prescribed without diurnal variations which are likely to impact $NO_3$ more than other oxidants. Here we estimate that including vertical profiles of emissions in Earth system models will increase the lifetime of NOx and hence the rate of NO3 production, but detailed studies are needed to assess this highly non-linear system; (3) chemical processes, with simplifications in the treatment of $N_2O_5$ heterogeneous chemistry being a potentially key area (as discussed by Archibald et al., (2020a) in the context of ozone budgets). As eluded to in Section 4, a key missing component in UKESM1 is the state-dependence of SOA formation. Whilst previous studies have used chamber studies to derive conditional state-dependence for the yield of SOA as a function of NOx (e.g., Pye et al. (2010)) future work should more holistically treat the suite of low volatility SOA precursors and their formation in the gas-phase and coupling to the aerosol-phase, as discussed by e.g., Schwantes et al. (2019); and (4) the chemical mechanism used, Archer-Nicholls et al., (2021) and Weber et al., (2021) show that the $NO_y$ budget is highly sensitive to use of a more comprehensive chemical mechanism in the UKESM1 framework. In this work we assessed the impacts of changing $k_{NO_3}$ (see the SI and S2 for details). We found that it is very difficult to predict how even a relatively simple change, like changing the rate constant for BVOC+$NO_3$ reactions, will modify the output of an Earth system model. Decreasing $k_{NO_3}$ results in vary spatially variable changes in $NO_3$, $NO_2$ and $O_3$. In spite of the large change in $k_{NO_3}$ the changes in the fluxes of BVOC+$NO_3$ oxidation were much more muted. Dedicated uncertainty analysis looking at points (1) to (4) above are needed. Finally, almost all CMIP6 models that have simulated interactive chemistry have simulated changes in $NO_3$ and we encourage the modelling community to undertake a multi model analyses similar to those focused on OH (Stevenson et al., 2020) and $O_3$ (Griffiths et al., 2021). Further multi model analyses are required to constrain the level of model uncertainty in predictions of $NO_3$ and the potential impacts this could have on future air quality and climate. Co-ordination and partnership between the $NO_3$ observational community and the modelling community could then allow the future changes in $NO_3$ to be observationally constrained.

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

**Code availability**

Due to intellectual property rights restrictions, we cannot provide either the source code or documentation papers for the UM. The Met Office Unified Model is available for use under licence. A number of research organisations and national meteorological services use the UM in collaboration with the Met Office to undertake basic atmospheric process research, produce forecasts, develop the UM code, and build and evaluate Earth system models. For further information on how to apply for a licence, see http://www.metoffice.gov.uk/research/modelling-systems/unified-model (last access: 1 October 2021).

**Data availability**

The UM data used to produce the figures are available from the Centre of Environmental Data Analysis (CEDA) (see SI for details, Archer-Nicholls et al., 2022).

**Author contribution**

ATA designed the research and supervised the analysis and led the revision of the manuscript. SAN and RA led the analysis. NLA and SAN performed the re-runs of the UKESM1 model. PTG contributed to the analysis. All authors contributed to writing the manuscript.

**Competing interests**

The authors declare that they have no conflict of interest.