# Peer review of "Large Simulated Future Changes in the Nitrate Radical Under the CMIP6 SSP Scenarios: Implications for Oxidation Chemistry"

_Atmospheric Chemistry and Physics, 2022_

## Author Comment (AC1)

Replies to referees "**Large Simulated Future Changes in the Nitrate Radical Under the CMIP6 SSP Scenarios: Implications for Oxidation Chemistry**" by Archer-Nicholls et al.

Note to Editor:
We would like to thank Dr. Tsigaridis for his time in editing our manuscript and bearing with us and the length of time it has taken us to reply to the reviewers comments. This was not a function of the amount of work involved but just a function of over-commitment of the team members with other tasks, and the impact of the first-author moving jobs out-side of academia. Below we highlight in red and blue text the referees comments and in black text our replies.

Referee #1:
Archer-Nicholls et al. report model results on historical trends and future projections in nitrate radical (NO3) abundance in the lower atmosphere at global scale. There is a focus on regional hot spots, especially South Asia, where both the historical trends and future projections show large differences. There is also a focus on the relevance of these changes for oxidation of biogenic volatile organic compounds (BVOC), which in turn are relevant to the efficiency with which these species produce secondary organic aerosol (SOA). The future projections are based on a set of emissions scenarios from the recent literature. Figure 2 shows the core result of the analysis of global maps of O3, NO2 and NO3 differences between the present day, preindustrial, and a series of future projections. Presuming that the O3 and NO2 differences are correct, the NO3 differences can be largely, although not fully, understood in terms of the changes in NOx emissions and their effects on O3 distributions.

The paper is of interest to ACP and publishable with minor revisions, as outlined below.

We would like to thank the referee for their time in reading our manuscript and their helpful comments that we have addressed and agree have improved the paper.

The most important comment, listed first below, is that the scope of the paper is somewhat limited compared to what it could be. The paper stops at mixing ratios and oxidation rates, without really predicting more about the associated changes in fates of BVOC oxidation products.

We understand that the referee would like the paper to have delved deeper but the simplicity of the simulation of SOA in the model means that some of the referees interesting points, i.e. change in BVOC SOA composition, are not simulated and so we can go no further with the analysis.

**Major Comments**

While the paper is of value in assessing trends in NO3 mixing ratio and BVOC oxidation rates, it stops short of assessing other important quantities such as organic nitrogen and SOA mass. For example, mass yields tend to be oxidant specific, and that effect is not captured here. Previous papers that have examined the mass yield dependences for SOA or organic nitrogen should be cited and compared to this model where possible. Relevant references are listed below.

1. von Kuhlmann et al., Sensitivities in global scale modeling of isoprene. Atmos. Chem. Phys., 2004. 4: p. 1-17.
2. Horowitz, L.W., et al., Observational constraints on the chemistry of isoprene nitrates over the eastern United States. J. Geophys. Res., 2007. 112(D12): p. D12S08.
3. Hoyle, C.R., et al., Anthropogenic influence on SOA and the resulting radiative forcing. Atmos. Chem. Phys., 2009. 9(8): p. 2715-2728.

4. Brown, S.S., et al, Nocturnal isoprene oxidation over the Northeast United States in summer and its impact on reactive nitrogen partitioning and secondary organic aerosol. Atmos. Chem. Phys., 2009. 9: p. 3027-3042.
5. Pye, H.O.T., et al., Global modeling of organic aerosol: the importance of reactive nitrogen (NOx and NO3). Atmos. Chem. Phys., 2010. 10(22): p. 11261-11276.
6. Hoyle, C.R., et al, A review of the anthropogenic influence on biogenic secondary organic aerosol. Atmos. Chem. Phys., 2011. 11: p. 321-343.
7. Schwantes, R.H., et al. , Comprehensive isoprene and terpene chemistry improves simulated surface ozone in the southeastern U.S. Atmos. Chem. Phys. Discuss., 2019. 2019: p. 1-52.

Our primary aim with this study was to quantify how [NO3] would change under future climate and emission scenarios and what the implications for the oxidising capacity of these changes would be. This is the first study of its kind to do this and we think sufficiently novel for publication. As an additional key aim, we documented these changes in a state-of-art Earth system model that has contributed significant data to CMIP6. By its nature, the model is a compromise of complexity and does not include the most complex representation of all processes. As the referee highlights, a number of schemes have been developed that modify the yield of SOA based on the environmental conditions under which SOA are generated (for example, the [NOx]). Our scheme does not do this, but it does allow us to calculate the fraction of monoterpenes oxidised via the NO3 pathway, which goes someway toward this point. We wholeheartedly agree with the referee that our study stops short and if we could go further with the analysis of, for example, changes in the composition of SOA, we would. We suggest this as the basis of further work and will modify the text accordingly as well as incorporating the key references to previous literature as highlighted by the referee.

We have added the following text to the Discussion section as we feel it is best placed there: "In addition, the simplistic treatment of SOA in UKESM1 prevents us being able to explore how the composition of SOA may change under the changing climate and emission scenarios that have been explored for CMIP6. Many previous studies have highlighted how the formation of SOA is highly sensitive to the conditions it is being formed under and how the composition of SOA will change concomitantly (e.g., Hoyle et al., 2011; Pye et al., 2010; Schwantes et al., 2019). These feedbacks between changes in SOA formation mechanism and SOA composition are being explored through future work in UKESM."

**Specific Comments**

Line 9: The nitrate radical is not always, or perhaps even in an integrated or average way, the principal oxidant during the night. This is more typically ozone. Figures within the paper show the importance of ozone compared to nitrate radical. Suggest rephrasing as either "principal oxidant together with ozone", or as "principal oxidant in areas with substantial NOx pollution."

We have made the corresponding change.

Line 24: Omit the word "rapidly". Reaction 1 is quite slow.

We have removed the word.

Line 28: See comment above from the abstract – need to qualify NO3 as most important nighttime oxidant since it always acts together with O3 and in locations without NOx emissions is an unimportant oxidant. A small but important caveat. See for example Edwards et al. Nature Geosci, 2017. 10(7): p. 490-495.

We have added in the key point about dependence on NOx.

Line 75: Worth noting here also that the rate constant for reaction (1) has among the strongest temperature dependence of any major atmospheric bimolecular reaction, so the source reaction is also sensitive to temperature increases.  This effect is certainly more modest than the N2O5 equilibrium but worth noting.

We have added some text which touches on this at line 37 after reactions are first introduced. "This temperature dependence in the equilibrium between $NO_2$, $NO_3$ and $N_2O_5$ is further driven by the extreme temperature dependence in the formation of $NO_3$ through R1."

Line 100: Agree with the caveats stated here that the simplification of a large, fixed uptake coefficient for N2O5 will affect the model predictions of various processes, including BVOC oxidation.  It would be useful to see a sensitivity test with a smaller uptake coefficient (e.g., 0.01 rather than 0.1 since the former is likely the more appropriate order of magnitude for the troposphere) for predictions regarding major process chemistry using the specific model in this paper rather than the reference to Jones et al.  The authors may wish to comment on the feasibility of inclusion of such a test, or at least qualitatively predict the outcome, if they elect not to do so.  See McDuffie et al. Journal of Geophysical Research: Atmospheres, 2018. 123(8): p. 4345-4372.  for a discussion of the complexity in the N2O5 uptake coefficient and its range of variability.

We have expanded on our original discussion on the likely impacts of the simplistic treatment of $N_2O_5$ uptake, and made reference to the McDuffie et al. (2018) study. "McDuffie et al. (2018) have determined a median g=0.0143 (range: $2x10^{-5}$ to 0.1751) based on constrained modelling of aircraft observations. Inclusion of this median uptake coeffecinet would be likely to lead to an increase in the atmospheric mixing ratio of $N_xO_y$ and thus our results can be seen as lower bounds on the potential changes in [$NO_3$]."

Line 134: Doubling of monoterpenes to account for isoprene is not clear.  It was stated above that isoprene is treated separately as its own species?

Yes, in the model mechanism isoprene is a unique species, but it does not feed into the SOA scheme. Instead, to account for the absence of isoprene oxidation forming SOA, the yield of SOA from monoterpene oxidation is doubled, as outlined in the cited papers (Mann et al., 2010, Mulcahy et al., 2020). In our more recent updates to the chemistry we now simulate isoprene derived SOA independently (e.g., Weber et al., 2022).

Line 139, 141: The expressions for kNO3 appear incomplete or else the units are other than expected for a bimolecular rate constant.  The prefactor should be a much smaller number if these units are in cm3 s-1.

Thank you for pointing this error out! It was a typographical error and has been fixed in the revised manuscript.

Line 159-161: There appear to be other features in the comparison of figure 1 for model measurement disagreement.  Most obvious is boundary layer height, and presumably vertical mixing throughout the model.  The NO2 gradient near the surface is very strong in the model but not as strong in the observations.  This is reflected in the ozone simulation as well.  The large NO3 at higher altitude relative to the model is also certainly a consequence of the NO2 at higher altitude, again something that could be attributed to vertical mixing that is too small (vertical gradients that are too large) in the model.

This is an excellent point and one we agree with. We have added the following text to reflect it. "In addition to the poor horizontal resolution models like UKESM1 also suffer in biases in vertical resolution and mixing. The simulation of boundary layer height in models like UKESM1 is difficult and

Figure 1 suggests that the model boundary layer height is much lower than the observations. This would make sense if in the model there is a significant land fraction in the grid boxed being analysed, as is the case."

Line 167-174 and Table 1: A useful comparison of model to observations for NO3. The authors state that these are all from surface observations. Related to the preceding comment, the vertical distribution is likely the most difficult aspect for a coarse resolution model, and even observations with small differences in elevation above surface might differ considerably in how accurately they are simulated. The authors may wish to add this caveat to the discussion.

We have added a comment to make this caveat clearer. "..., and in light of the caveats already discussed through the analysis of Figure 1, we find that…"

Figure 3: Useful here would be to also plot absolute temperature across the top axis to provide the reader an easy reference to the temperature changes that are actually inferred by the models. Similarly, rather than a natural logarithm, a base 10 log on the y axis would make the translation of the equilibrium ratios easier to understand at glance rather than having to invert an exponential function.

We have corrected the Figure 3 as suggested.

Line 260-262: The choice of presentation using rates is somewhat misleading since it is an average rate over a diel cycle and a month. An integral (i.e., a total mass within a given time period) would be a more appropriate quantity in figures 4 and 5. The figures themselves would presumably not change, but the mass would place the figures in better context for emissions inventories of BVOC, which are typically in mass units rather than rates.

We disagree on this point. We feel that the presentation of rates is instructive as is.

Line 297: The caveat about diel boundary layer variability is almost certainly not limited to East Asia, as implied.

Agreed and corrected.

**Technical corrections**

Line 99: Its rather than it's

Corrected.
Line 122: -pinene is missing either an alpha or a beta, likely.

Corrected.
Figure 1c: NO3 is given in ppbv when pptv is almost certainly what was intended.

Corrected.
Line 295: No comma after the word include

Corrected.

References cited here but not in the original text:
Weber, J., Archer-Nicholls, S., Abraham, N.L., Shin, Y.M., Griffiths, P., Grosvenor, D.P., Scott, C.E. and Archibald, A.T., 2022. Chemistry-driven changes strongly influence climate forcing from vegetation emissions. Nature Communications, 13(1), p.7202.

Referee #2:
The study discusses the evolution of NO3 radicals from 1850-2100 based on model simulations by UKEMS1 Earth System Model under different climatic scenarios. Special attention is paid to South Asia where NO3 is expected to increase to unprecedented levels. In general, the study is well established and well written. A few comments are listed as follows for the authors to consider.

Again, we would like to thank the referee for their time in reading our manuscript and their helpful comments that we have addressed and agree have improved the paper.

1. Abstract: It is expected that some quantitative results be mentioned in the abstract, instead of using vague descriptions like "dramatic increase", "rapid growth" and "sharp decline".

We agree and have added in some more quantitative statements to improve the abstract.

2. Fig. 1, (1) The legend and lines in Fig.1 overlap which needs to be modified later. (2)The upper-limit of the x-axis may be larger, in order to allow the maxima concentration (the upper limit of the error bar) be included in the figure (The same suggestion for Fig.S4).

We have modified the Figures as suggested by the referee.

3. Fig. 3: The x-axis could be changed into fractions (i.e. 1/298).

We have modified the figure in agreement with referees comments too.

4. Fig. 5: The y-axis could be changed into log scale in order to make the zonal distribution clearer. (Same suggestion for Fig. S5)

We have modified the Figure 5 and S5.

5. Discussion: The discussion part is too short. It is suggested that discussion could be incorporated into results.

Rather than adding in the results, which we have summarised in the conclusions, we felt it was an appropriate point to raise the major weakness suggested by referee 1. In adding in this we feel we have expanded on the discussion.

6. An uncertainty analysis of the simulations (or at least an uncertainty analysis of the kinetics parameters) is required in the article or supplement. What is the most important cause of model uncertainty? And how will it influence the model results?

A full uncertainty analysis is beyond the scope of this paper but we have expanded the discussion of key uncertainties at line 316 onwards to include a comment on what we expect the impacts of each uncertainty to be on the model fields.

One key uncertainty we have assessed is the kinetics of BVOCs with $NO_3$, which we found to have a modest impact on the results (through updating the $kNO_3$). What is clear is that the back-of-envelope predictions of the impact of a change are not accurate (at least in the case for $kNO_3$). We changed $kNO_3$ by roughly a factor of 4 and the changes in oxidation rates of BVOCs by NO3 were much more modest (reductions of 5-30%, Figures S2 and S3) owing to the concomitant changes in the concentrations of the key oxidants (Figure S1).

7. The model results demonstrate that NO3 levels may double by the end of 21st Its further implications could be discussed in depth in the conclusion part. Does it mean more BVOCs oxidized by NO3 and more SOA or particulate nitrate production in the future and so what?

These are good points and we have expanded the Conclusions section to make the case clearer for why this work is important. "The impacts of an increase in $NO_3$ dominated BVOC oxidation are, as yet, uncertain. An increasing body of literature is examining the mechanistic pathways through which BVOCs and $NO_3$ react and the impacts of BVOC+$NO_3$ derived SOA. He et al. (2021) have shown through detailed laboratory experiments using cavity enhanced absorption spectroscopy that hat some of the organic nitrates in BVOC+$NO_3$ derived SOA may serve as atmosphere-stable NOx sinks, or reservoirs, and will absorb and scatter incoming solar radiation during the daytime leading to an anthropogenic radiative forcing component (given that $NO_3$ is primarily an anthropogenic species)."